# STEERABLE VIDEO ACTION MODEL

## ABSTRACT

Steerable robot policies—those conditioned on steering signals like trajectory traces—offer a promising solution for flexible, general-purpose robot control. However, most existing steerable policies are limited by their reliance on action-labeled robot data for learning to follow these steering signals. The recently proposed video-action models offer a scalable solution for incorporating additional video data by learning to jointly predict future video frames along with actions, which enables the learning of rich latent representations that capture visual dynamics and helps improve action prediction. Despite their promise, prior video-action models are not steerable, limiting their ability to generalize to out-of-distribution task specifications or novel object configurations that require new behaviors. We propose the Steerable Video Action (SVA) model, which learns to jointly predict future video frames and low-level actions while receiving guidance from end-effector trajectory traces as steering signals. To process these traces, we represent them as images, encode them using a pretrained VAE, and explicitly align the encoded tokens spatially with visual observation tokens before passing them through a transformer. We find that SVA can incorporate guidance from end-effector trajectory traces and generalize better to unseen traces outperforming baselines with and without access to trajectory traces.

## 1 INTRODUCTION

Steerable robot policies—those that are conditioned on steering signals like trajectories (Gu et al., 2024; Zheng et al., 2025; Lee et al., 2025), keypoints (Sundaresan et al., 2023), or language corrections (Belkhale et al., 2024; Shi et al., 2025)—enable flexible behaviors across a wide range of tasks by simply varying the steering signal. These steering signals can come directly from human users or be automatically generated given the high-level task instruction along with an initial image of the scene using state-of-the-art vision-language models (VLMs) (Team et al., 2025).

While promising, most existing steerable policies are constrained by how

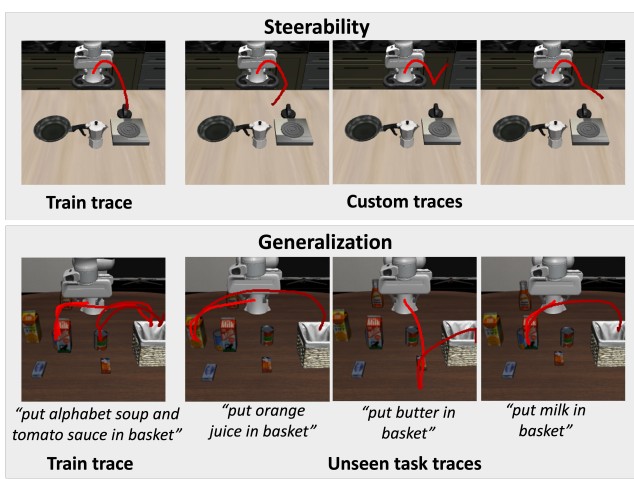

"put alphabet soup and tomato sauce in basket"    "put orange juice in basket"    "put butter in basket"    "put milk in basket"

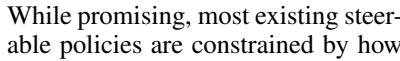

Figure 1: **Steerability and Generalization.** Conditioning on steering signals, like end-effector traces offers Steerable Video Action model (SVA) steerability and generalization.

they are trained: they rely heavily on action-labeled robot data for learning to follow the steering signals. This makes them expensive to scale and brittle when deployed in new tasks, environments, or embodiments. Because these models are tightly coupled to their training distribution, even modest shifts—such as unseen object configurations or out-of-distribution steering signals—can affect their steerability causing the task performance to degrade. As a result, their generalization remains limited, despite the flexibility that steerability aims to provide.

A key question is how to train steerable policies without relying solely on costly, embodiment-specific robot data. One promising direction is to use video supervision by leveraging large-scale video datasets that capture rich physical interactions across diverse environments. Recent video-action models (Guo et al., 2024; Zhu et al., 2025), which generate videos and actions simultaneously, demonstrate the benefits of video prediction-based pretraining for improving action prediction. However, these models are not designed to be steerable and are typically trained to solve specific tasks, limiting their ability to generalize to out-of-distribution task specifications or novel object configurations that require new behaviors.

In this work, we study the problem of incorporating steering signals into video action models. We introduce the Steerable Video Action model (SVA) model, which takes as input a history of observations and a steering signal, an end-effector trace the robot needs to follow, and predicts future observations and actions that follow the given trace. Unlike previous steerable action models such as the RT-Trajectory (Gu et al., 2024; Lee et al., 2025), our approach jointly predicts future video frames and actions. This allows the model to learn a rich latent representation of visual dynamics that enhances action prediction. Moreover, our framework can leverage large amounts of actionless video data for pretraining, enabling the model to acquire diverse behaviors guided by steering signals without relying on explicit robot actions. This pretraining establishes a strong prior for joint video-action learning, improving both sample efficiency and generalization to novel tasks.

A natural question that arises is: *how should steering signals be incorporated into video–action models so as to improve both steerability and generalization?* In this work, we study this by using end-effector traces as the primary form of guidance. We find that representing traces as images and explicitly aligning the encoded tokens between trace and observation images is crucial for capturing the spatial correspondences required for accurate trace following. This alignment mechanism enables coherent prediction of future observations and actions that adhere to the intended trajectory. Our experiments show that SVA provides more precise and flexible control, outperforming the strongest baseline by 33%.

To summarize, our work has four main contributions.

- First, we introduce the Steerable Video Action model, which enables flexible control by conditioning on steering signals, specifically end-effector traces. By jointly predicting future video frames and actions, SVA learns to align its behavior with the visual trajectory implied by the steering signal, resulting in more accurate execution of steering inputs.

- Second, we show that SVA generalizes effectively to settings that require following unseen traces. SVA achieves an improvement of 33% over the strongest baseline on unseen task traces.

- Third, SVA supports pretraining on actionless videos, allowing the model to learn from diverse physical interactions and expand the range of steering signals it can reliably follow without requiring action labels, further improving steerability and generalization.

- Finally, we analyze how to incorporate steering signals into video-action models, showing that spatial alignment between trace and observation tokens, is critical for accurate trace following.

## 2 RELATED WORK

In this section, we begin by reviewing prior works using steering robot policies. We then discuss video generation for robotic control. Finally, we examine previous works that utilize action-free video data for robot learning.

**Steerable robot policies.** Numerous prior works have attempted to develop steerable robot policies that can be controlled via trajectories (Li et al., 2025b; Zheng et al., 2025; Gu et al., 2024), keypoints (Sundaresan et al., 2023) or low-level language motions (Belkhale et al., 2024; Shi et al., 2025). These methods typically use a separate low-level policy (Li et al., 2025b) or a separate prediction stage (Lee et al., 2025) to output actions that follow steering signals using action data from a single robot embodiment. However, since data for a single embodiment is often limited, these approaches struggle to generalize to tasks beyond the training tasks they were trained on. In contrast, the action prediction in our model can benefit from action-free data, as it shares the same latent representation as the video prediction head, thereby improving generalization.

**Steerable Video Generation.** A parallel line of research has solely looked into controllable video generation using object trajectories (Yin et al., 2023; Wang et al., 2023; Wu et al., 2024; Wang et al., 2025), edge map videos (Zhang et al., 2025) and camera trajectories (He et al., 2025; NVIDIA et al., 2025) control the generations. These methods steer synthesis by converting the user's motion cues into explicit constraints on the diffusion process: DragNUWA (Yin et al., 2023) and MotionCtrl (Wang et al., 2023) rasterize object paths (or optical-flow targets) into conditioning masks, DragAnything (Wu et al., 2024) warps the latent feature grid around dragged keypoints, ATI (Wang et al., 2025) injects hierarchical velocity tokens, Zhang et al. (Zhang et al., 2025) feed edge-map sequences as pixel-wise guidance, and CameraCtrl (He et al., 2025) applies per-frame pose-delta matrices to align the generated viewpoint. However, these approaches mostly target non-embodied settings. Cosmos (NVIDIA et al., 2025) loosely explores controllable video generation in an embodied setting of autonomous driving using trajectories to control the vehicle. However, in this work, we explore steerability in the robot manipulation setting which requires more fine-grained control.

**Video Generation for Robotics.** Numerous robotics works have explored leveraging recent advancements in image and video generation, using them for task planning (Du et al., 2023; 2024), generating augmentations (Chen et al., 0; Yu et al., 2023; Zhang et al., 2024) and producing visual chain-of-thoughts (Zhao et al., 2025). Pushing further, recent works (Li et al., 2025a; Zhu et al., 2025) have also sought to jointly model and predict both videos and actions, enabling a wide range of capabilities within a single model: standalone video and action prediction, forward dynamics and inverse dynamics. However, these models are limited in terms of the steerability of generated video and actions, typically relying on high-level instruction for task specification. This&That (Wang et al., 2024) controls robot motions in generated videos using gestures and language. However, its steering representation is limited to pointing at objects or static locations and cannot convey trajectory-level guidance. In our work, we attempt to incorporate fine-grained control in the form of trajectory sketches into video action models.

# 3 STEERING VIDEO AND POLICY GENERATION

We refer to *steerable* robot policies as policies that follow language instructions or other high-level representations of goals such as trajectories drawn on an image referring to the path the robot needs to take. Such policies (Gu et al., 2024; Belkhale et al., 2024; Lee et al., 2025) provide a steerable interface that allows users to effectively control the robot. In this section, we introduce Steerable Video Action model (SVA), a unified architecture that learns to follow diverse steering signals while jointly predicting future video frames and low-level robot actions.

**Problem Setup**. Given a sequence of past observations $\{\mathbf{o}_{t-h'+1}, \ldots, \mathbf{o}_t\}$ and steering signals $\{\mathbf{s}_{t+1}, \ldots, \mathbf{s}_{t+h}\}$, the goal is to generate future observations $\{\mathbf{o}_{t+1}, \ldots, \mathbf{o}_{t+h}\}$ and low-level actions $\{\mathbf{a}_t, \ldots, \mathbf{a}_{t+h-1}\}$ that are consistent with the provided steering inputs.

## 3.1 UNIFIED VIDEO ACTION MODEL

Video–action models (Li et al., 2025a; Zhu et al., 2025) have shown promise in improving policy inference. A recent method, the Unified Video Action (UVA) model (Li et al., 2025a), learns a joint video–action representation that predicts sequences of future observations and actions. During training, it leverages video supervision to learn effective vision–action representations, and at inference time it supports real-time action predictions by omitting video generation. We build SVA on top of UVA to enable steerability while maintaining its efficient joint prediction capabilities.

In UVA, each visual observation $\mathbf{o}_t$ is encoded using a pretrained VAE encoder, producing a sequence of latent representations of images $\{\mathbf{v}_1^t, \ldots, \mathbf{v}_N^t\}$. Similarly, the history of actions is encoded into a sequence of action tokens $\{\mathbf{a}_1^t, \ldots, \mathbf{a}_N^t\}$. These action tokens are concatenated with the latent representations of images along the channel dimension and passed through a transformer encoder, which fuses the visual and action information into a unified video–action representation. This joint latent representation $\{\mathbf{z}_1^t, \ldots, \mathbf{z}_N^t\}$ is then used to condition the diffusion-based decoders for generating video and actions.

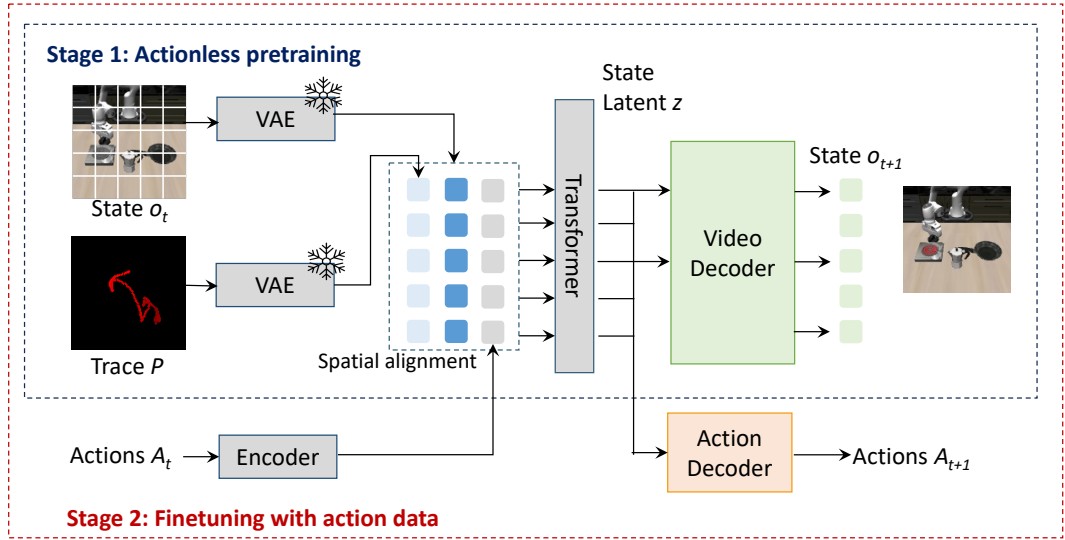

Figure 2: **Approach.** SVA conditions on traces by representing them as images, encoding them using a pretrained VAE encoder and concatenating them spatially with the tokens of context images.

## 3.2 STEERABLE VIDEO ACTION MODEL

While steering a policy can be done via different high-level signals such as language, sketches, keypoints, and others, we use 2D end-effector trajectories – referred to as robot traces – as our primary steering signal. Each trace is represented as a sequence of 2D coordinates $\mathbf{s}_t = (x_t, y_t)$, indicating the desired motion of the robot's end-effector over time.

**Trace Extraction.** The 2D traces can be extracted in different ways depending on the data source. For simulation data, we have access to both the 3D pose of the robot's end effector and the camera pose, which allows us to project the 3D positions into 2D space. We also incorporate additional real human data from (Clark et al., 2025) to pretrain the model for trace-conditioned video generation. The dataset contains 2000+ human demonstrations that mimic how a robot interacts with objects in tabletop manipulation tasks. Human hand trace are extracted using off-the-shelf hand trackers (Pavlakos et al., 2024).

**Trace Sampling.** To increase the model's flexibility in handling diverse trace patterns, we do not directly use the full trajectory trace. Instead, we expose the model to a wide range of trace lengths during training—from short motions to full task trajectories. Specifically, the trace start point is sampled from the interval $[0, t-h'+1]$, and the end point is sampled from $[t+h, T]$, where $T$ denotes the full length of the trajectory. This ensures that the trace spans both the historical observations fed to the model and the future observations to be predicted, providing temporal context that helps the model align past movements with future outcomes and better learn the dynamics of the task.

**Spatial Alignment.** To make trace signals more explicit and spatially aligned with visual observations, we render each trajectory as an image, encoding time via a color gradient. To indicate motion direction, we apply a linear intensity fade from the start to the end of the trajectory—where the starting point $P_s$ has maximum red channel intensity (value 1), the endpoint $P_e$ has reduced intensity (value 0.5), and intermediate points fade linearly from 1 to 0.5. This visual encoding helps the model associate spatial locations with temporal dynamics, as illustrated in Figure 2.

The trace image is passed through a pretrained VAE encoder, producing a sequence of latent tokens $\{\mathbf{p}_1^t, \ldots, \mathbf{p}_N^t\}$ for each timestep $t$. In parallel, each observation $\mathbf{o}_t$ is encoded into visual latents $\{\mathbf{v}_1^t, \ldots, \mathbf{v}_N^t\}$. The trace latents and visual latents are concatenated along the channel dimension at each spatial location, forming a fused representation. This combined input is passed to the transformer to jointly predict future video frames and corresponding robot actions. This de-

sign encourages the model to learn spatial and temporal correspondences between the trajectory and visual context, improving alignment between user intent and generated behavior.

**Video and Action Decoding.** Similar to UVA, we use diffusion decoders to predict future videos and actions. The decoders are trained using a Denoising Diffusion Probabilistic Model (DDPM) objective (Ho et al., 2020). During inference, the decoders progressively denoise random Gaussian noise to generate video and action sequences, respectively.

**Inference.** At test time, the model receives a steering signal in the form of a 2D trace (e.g., hand-drawn or extracted from another demonstration), along with a short context of past visual observations. It then generates a sequence of future video frames and the corresponding low-level robot actions, guided by the steering signal.

### 3.3 VARIANTS OF TRACE ENCODING

In addition to our proposed method, we explore alternative trace encoding strategies to investigate how to most effectively incorporate steering signals into video action models. Specifically, we evaluate four variants of our approach to study the influence of i) spatial alignment and ii) trace length variability

*SVA No Spatial Alignment:* We study the influence of spatial alignment between the trace image and the observation image. Instead of concatenating their latent token sequences along the channel dimension, as shown in Figure 2, we concatenate them along the sequence dimension, resulting in a sequence of $2N$ tokens. This removes the spatial alignment between trace and visual features before passing the sequence to the Transformer.

*SVA Fixed-Length Trace:* To examine the effect of trace length variability, we constrain the trace to a fixed interval $[t - h' + 1, t + h]$, covering only the immediate history and prediction horizon. Unlike the variable-length trace used in SVA, this variant removes randomness in trace span, allowing us to assess whether such variability is important for effective guidance.

### 3.4 CO-TRAINING WITH ACTIONLESS DATA

Similar to previous video-action policies (Ye et al., 2024), SVA benefits from large-scale video pretraining, which provides strong visual and temporal priors. In addition, exposure to diverse steering signals during pretraining enables the model to learn a wide range of behaviors without relying on robot action labels.

To achieve this, we first **pretrain** our model on a combination of action-free human video datasets and robot datasets using only the video generation objective. The model takes trace of human hand or robot gripper along with historical observations as input and predicts future observations. The 2D hand trajectories are extracted from human activity videos using a pose tracking method (Pavlakos et al., 2024), and serve as steering signals to guide the prediction. We then **finetune** the model by including a smaller set of robot demonstrations. During this stage, the model takes steering signals as input and is jointly optimized using both video generation and action prediction losses. This two-stage training process enables the model to retain generalization capabilities from large-scale video data while adapting to the specific action space of the robot.

## 4 EXPERIMENTS AND RESULTS

In this section, we present experiments to evaluate the effectiveness of our Steerable Video Action model (SVA) in terms of steerability (section 4.2) and generalization (section 4.3). We also assess the benefits of incorporating human video data (section 4.4) and finally investigate how different trace encoding mechanisms (section 4.5) influence performance in terms of steerability and generalization. We start by describing the steerability and generalization evaluations in the following.

- **Steerability:** We evaluate the ability of our model and its variants to follow a given steering signal—in our case, a 2D trace input extracted from unseen trajectories. All models take the steering trace as input and generate future video frames and actions. We use Fréchet Video Distance (Unterthiner et al. (2019)), or FVD, to measure video steering ability by comparing

generated videos against groundtruth videos. For evaluating action steering, we measure the mean squared error (MSE) between groundtruth actions and the actions realized by the models while attempting to follow the extracted traces.

- **Generalization:** The strong steerability of our model enables more accurate trace following, which in turn allows it to better complete tasks when guided by new traces. We evaluate the model's ability to generalize to previously unseen traces that are either extracted from held out task demonstrations or manually annotated by a human. Generalization performance is assessed by measuring the success rate of task completion under these novel traces.

## 4.1 Experimental Settings

We evaluate our method in simulated robot environments from the LIBERO benchmark (Liu et al., 2023), focusing on how joint video–action modeling and actionless data improve steerability and generalization. Below, we describe the datasets, tasks, and baselines used in our experiments.

**Data.** Our experiments combine robot demonstrations and actionless human videos to study three questions: (i) whether modeling videos and actions together improves learning, (ii) whether incorporating actionless task data benefits generalization, and (iii) whether human data from a different embodiment improves trace-following.

- **LIBERO-90 without actions.** We use the LIBERO-90 benchmark (90 tabletop manipulation tasks across 20 environments) without actions to test whether actionless data contributes to solving novel tasks. Each task provides 45 demonstrations for training and 5 held-out demonstrations for evaluation. Actionless LIBERO-90 data is included in both training stages.
- **Human video dataset.** To evaluate cross-embodiment benefits, we use 2,416 episodes of a human hand performing tasks in a toy sink environment with a fixed third-person camera (Clark et al., 2025). This controlled setup avoids camera-motion variability common in human datasets. Since many LIBERO tasks are pick-and-place, including this dataset allows us to test whether human traces improve steering for similar tasks. We use this data only in first stage of training. The results in the paper do not use this data unless specified.
- **LIBERO-10 with actions.** We use LIBERO-10 as our sole source of action supervision. This benchmark includes 10 longer-horizon tasks. We use 45 demonstrations per task for training.

**Tasks.** All evaluations are conducted in the LIBERO framework. For LIBERO-90 tasks, we hold out five demonstrations per task as ground-truth traces. These traces are used both to evaluate trace-following ability and to measure whether actionless data improves performance on unseen tasks. We further test qualitative generalization to custom instructions that involve novel objects or modified spatial goals.

**Baselines.** We compare our method (SVA) against four baselines, covering both non-steerable and steerable models. All baselines are trained or finetuned on the same data as SVA.

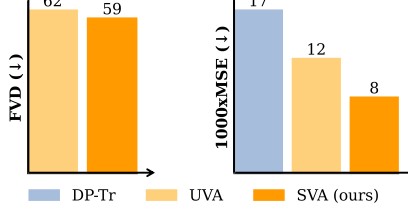

- **Unified Video Action Model (UVA)** (Li et al., 2025a): A state-of-the-art joint video–action model trained without conditioning on steering traces.
- **Diffusion Policy (DP)** (Chi et al., 2023): A diffusion-based action policy conditioned on language instructions, with CLIP encoders for language grounding.
- **DP-Tr**: An extension of Diffusion Policy that conditions on 2D steering traces, enabling trace-guided action generation.

Figure 3: **Video and policy steering.** SVA generates videos and actions that are more closer to the groundtruth trajectories compared to the baselines.

## 4.2 Can SVA effectively be steered by traces?

We quantitatively evaluate the trace-following ability of our method and baselines using held-out demonstrations from LIBERO-90. Results are summarized

in Figure 3. Our approach, which explicitly conditions on traces, produces videos that better align with ground-truth rollouts, achieving significantly lower Fréchet Video Distance (FVD) compared to the non-steerable UVA baseline. This indicates that trace conditioning provides a strong supervisory signal for generating temporally coherent, trace-aligned visual predictions. As shown in Figure 3, SVA not only matches the ground-truth traces visually but also produces action sequences that are quantitatively closer to the demonstrated actions. In particular, the mean squared error (MSE) between predicted and ground-truth actions is reduced by $52\%$ relative to DP-Tr, a steerable baseline.

**Steering with LIBERO-90 task traces**

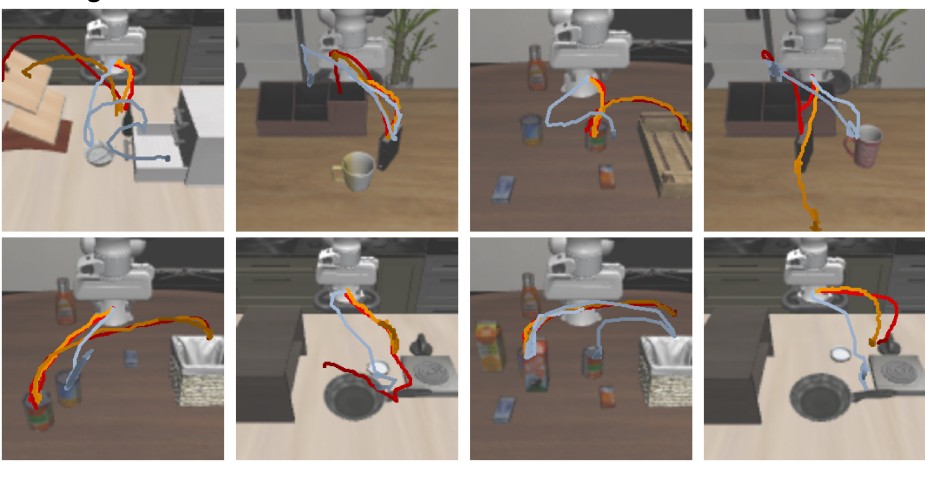

**Steering with human-drawn traces**

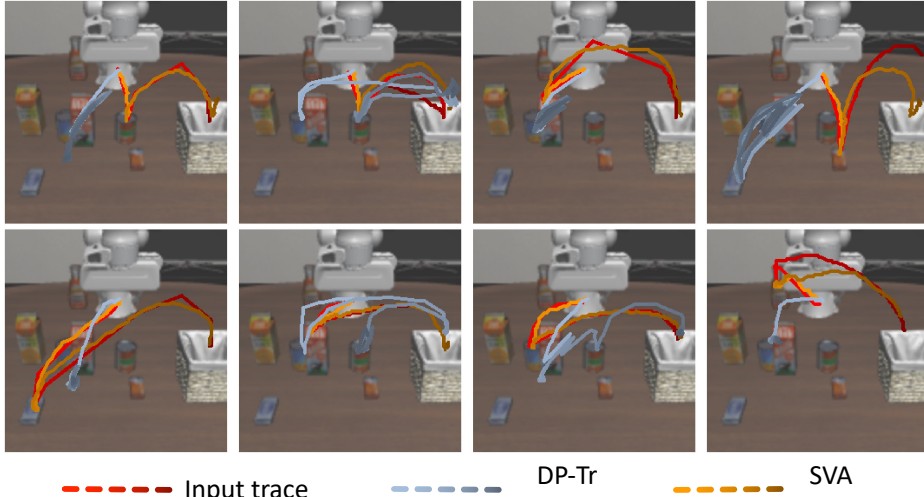

- - - - - Input trace        - - - - - DP-Tr (baseline)        - - - - - SVA (ours)

Figure 4: **Policy steering (qualitative).** We extract traces from task demonstrations (top rows) and use custom human sketches (bottom rows) to condition steerable policies. The results indicate that SVA more closely follows the input traces (red) compared to DP-Tr.

We also observe clear qualitative differences. As illustrated in Figure 4, SVA is able to follow traces extracted from demonstrations with high spatial fidelity, while also generalizing to free-form traces drawn by *humans*. In these settings, DP-Tr often deviates from the intended trajectory, whereas SVA produces rollouts that are closely aligned with the input traces.

### 4.3 CAN SVA GENERALIZE TO UNSEEN TASK TRACES?

We compare SVA against both steerable and non-steerable baselines on LIBERO-90 tasks for which no action data was available during training. As shown in Figure 5, SVA demonstrates substantially

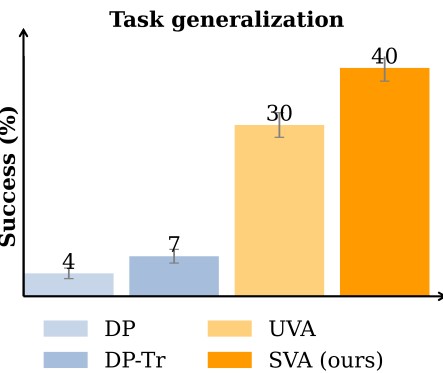

Figure 5: **Generalization to unseen traces.** When evaluated on held out demonstrations from LIBERO-90, SVA outperforms steerable and non-steerable policies.

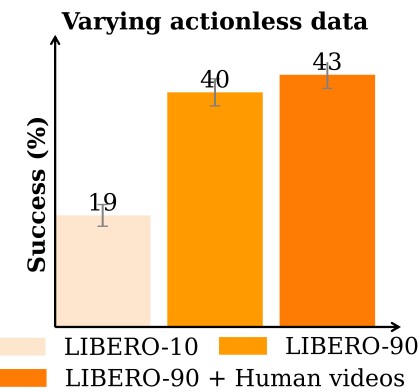

Figure 6: **Varying actionless data.** SVA benefits from actionless LIBERO-90 data and can further improve by additional pretraining with human video data (Clark et al., 2025).

stronger generalization to unseen traces than all baselines. In particular, it outperforms the state-of-the-art UVA model, which lacks steering inputs, and achieves an absolute improvement of 33% over the steerable DP-Tr policy. DP-Tr, which is only trained on action data from LIBERO-10, fails to adapt to new task traces (fig. 4). These results underscore the importance of joint video–action modeling for learning steerable policies, especially in settings where action supervision is unavailable for many tasks.

## 4.4 CAN SVA BENEFIT FROM ACTIONLESS HUMAN VIDEOS?

Our main experiments use LIBERO-10 action data, supplemented with action-free videos from LIBERO-90. To assess the role of this additional supervision, we compare three settings: (i) pretraining with only LIBERO-10 videos, (ii) pretraining with LIBERO-10 actions plus actionless LIBERO-90 videos, and (iii) further enriching this setup with human videos Clark et al. (2025). We find that restricting actionless pretraining to the smaller and less diverse LIBERO-10 data degrades performance by $-21\%$, whereas augmenting with human videos improves generalization by $+3\%$ on unseen traces. These results highlight that diverse, actionless human videos provide a valuable signal for improving steerability in novel tasks, and suggest a promising path toward scaling robot learning by leveraging large, unlabeled human video corpora readily available in the wild.

## 4.5 HOW TO EFFECTIVELY CONDITION ON TRACES?

We investigate alternative ways of incorporating steering traces into SVA. Specifically, we compare (i) fixed-length traces restricted to the prediction context window, and (ii) concatenating trace tokens directly to the context tokens without enforcing spatial alignment. All variants are trained for the same number of steps in both pretraining and finetuning stages, ensuring a fair comparison.

To evaluate these design choices, we sample random trace lengths that extend beyond the prediction context and report FVD scores on 256 held-out windows from LIBERO-90 demonstrations. As shown in fig. 7, restricting traces to the context window significantly reduces performance, underscoring the importance of conditioning on longer traces that provide guidance beyond the immediate horizon. Moreover, simply concatenating trace tokens without alignment degrades results. By spatially aligning trace tokens with observation tokens, the model can exploit consistent correspondences between the desired end-effector positions and visual scene structure. This alignment is crucial for accurate trace following, enabling coherent

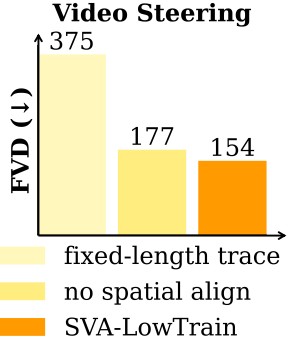

Figure 7: **Ablations on trace conditioning.** We experiment with different ways of representing traces and conditioning on them.

video rollouts and action predictions that adhere to the intended trajectory. Taken together, these findings demonstrate that both trace length randomization and spatial alignment are essential design choices for effective steering.

## 5 CONCLUSION

In summary, we present Steerable Video Action model (SVA), a video-action model that can jointly forecast future video frames and low-level robot actions while accepting control inputs in the form of end-effector traces. This unified design yields precise, user-controllable behaviors and surpasses prior controllable and non-controllable baselines for following unseen traces. Finally, by absorbing large-scale action-free human video, SVA has the potential to learn a richer latent representation that can further boost policy performance. Beyond improved task execution, SVA highlights how trace conditioning can transform video–action models into more flexible and general-purpose steerable policies.

A central strength of our approach lies in its ability to absorb large-scale action-free video, including human demonstrations, to establish a rich prior over physical interactions. This enables effective steering without requiring dense action labels and suggests a practical pathway to scaling robot learning with widely available video data. Our analysis also underscores the importance of spatial alignment between trace and observation representations, offering insights for designing future steerable models.

Looking ahead, SVA offers a foundation for more intuitive and scalable robot control. Extending steering inputs beyond traces—to keypoints, free-form language, or multimodal human feedback—could enable richer and more natural interaction between humans and robots. Another promising direction is leveraging diverse data from humans and multiple robot embodiments to train more general steerable policies, ultimately supporting deployment on real-world robot platforms.

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
