# OpenReview forum: "Steerable Video Action Model"
_ICLR.cc/2026/Conference — ICLR 2026 Conference Withdrawn Submission_

### Official Review · Reviewer_JAg4 · 2025-10-21

**Soundness:** 2
**Presentation:** 2
**Contribution:** 1
**Rating:** 2
**Confidence:** 4

**Summary:**

The paper proposes Steerable Video-Action (SVA) model that conditions the joint video+action backbone from UVA [1] on 2D end-effector traces rendered as images. Traces are encoded and spatially aligned with visual tokens; diffusion heads predict future video and low-level actions. The authors claim better trace following and higher task success on LIBERO tasks, plus benefits from pretraining with action-free videos.

[1] Unified Video Action Model

**Strengths:**

- A simple interface that can condition the rendered end-effector traces as images with the video-action backbone
- Explores different mechanism for encoding and conditioning traces

**Weaknesses:**

- Limited novelty. The only new component is how traces are injected (rendered 2D trajectories + spatially aligned conditioning).
- All results are on LIBERO. These environments share strongly overlapping motion primitives; success on a small held-out set that is distributionally similar to training does not demonstrate robustness. There are no random seed sweeps, no domain shifts (camera/viewpoint/object layouts), and no evidence it would transfer to real robots. Critically, the paper does not test out-of-distribution traces (e.g., shapes and kinematic demands that differ substantially from GT actions). As a result, the central claim of steerability that generalizes to unseen traces is not supported.
- Trace vs. pretraining. A strong alternative is to use extracted end-effector poses from video to pretrain a pseudo-policy and then fine-tune a downstream controller, similar to EgoVLA [1]. The paper does not compare against this line, yet this is the most pertinent question: is explicit trace conditioning better than simply pretraining on trajectories and then fine-tuning?
- It is well established that UVA-like setup often drive most of the gains from the video prediction objective. But the paper doesn’t compare fairly with other diffusion-based video prediction approaches (e.g., recent unified or predictive models such as UniPI [2], PAD [3], etc.). If video prediction is the main driver, SVA’s marginal benefit from trace conditioning must be isolated and measured.
- The paper reports FVD and action MSE. FVD is a weak proxy for path adherence, and the reported differences between UVA and SVA are small.


Overall, the architectural change (trace rendering + conditioning) appears incremental, and the current experiments do not convincingly establish the promised steerability or generalization beyond LIBERO. Core baselines are missing, evaluation metrics do not directly measure trace adherence, and the trace dataset itself is insufficiently specified. With stronger, trace-faithful metrics, out-of-distribution tests, and more related baselines comparison, the work could become more compelling.

[1] EgoVLA: Learning Vision-Language-Action Models from Egocentric Human Videos

[2] Learning Universal Policies via Text-Guided Video Generation

[3] Prediction with Action: Visual Policy Learning via Joint Denoising Process

**Questions:**

- How are evaluation traces generated and how different are they from ground-truth action paths? Some form of quantitative analysis here would be insightful.
- What happens if you remove trace conditioning and instead pretrain on extracted end-effector trajectories to initialize a downstream policy, which is then finetuned. Does SVA still win on task success/trace following?
- Comparison with other diffusion-based video prediction approaches. More experiments to clearly distinguish the gains due to trace conditioning from video prediction objectives.

---

### Official Review · Reviewer_9SuK · 2025-10-27

**Soundness:** 2
**Presentation:** 2
**Contribution:** 2
**Rating:** 2
**Confidence:** 3

**Summary:**

This paper extends video–action models to the steerable setting by introducing 2D end-effector trace signals as a condition. The traces are rendered as images, encoded with a VAE, spatially aligned with visual tokens, and processed in a transformer; a diffusion head then predicts future frames and low-level actions. The authors claim that this approach improves trace-following, with a reported 33% improvement on unseen traces, and that pretraining with “actionless” human videos further enhances generalization.

**Strengths:**

The steerability objective is intuitive and practically relevant. The proposed integration of traces into the UVA framework is straightforward.

**Weaknesses:**

1.	Limited contribution: The method is essentially a trivial extension of UVA by adding an extra modality (trace as a condition). Since traces themselves are a kind of action representations, the method could be seen as combining two action spaces in UVA, rather than introducing a fundamentally new idea.
2.	Definition issues: Related to the above, the labeling of traces often depends on underlying action annotations (e.g., end effector pose in Libero, hand pose from HaMeR), considering some recent works[1,2,3]. If so, can the pretraining really be considered “actionless”?
3.	Insufficient experiments: Performance improvements are shown only in LIBERO, with no results in other simulators, no real-robot validation, and a lack of strong baselines. Experimental details are unclear:
    - How are DP and UVA trained without action or trace inputs?
    - How exactly is DP_tr implemented? Are RGB and trace signals aligned spatially?
    - In Fig. 4, DP_tr  almost entirely fails to follow signals, raising concerns about baseline fairness.
    - It is unclear how FVD is computed and to what extent it reflects steerability.
    - Using only LIBERO-90 and LIBERO-10 raises questions: how different are these distributions really?
    - Some potential baselines (RT-Trajectory[4], TraceVLA[5]) are omitted.
4.	Design choices under-explained:
    - Why represent steerable signals as 2D trace images instead of keypoints or other alternatives?
    - Would alternative trace encodings affect performance?
    - Does the history window length affect performance?
    - How does the ratio of “actionless” data impact results?
    - What about the recipes for libero data without action and human data without action?
5.	Limited scalability: The spatial alignment design requires a fixed camera for consistent correspondence, whereas most human “actionless” data (e.g., Ego4D[6], Epic-Kitchens[7]) comes from wearable cameras. Does this explain why such datasets were not used here?

[1] [EgoVLA : Learning Vision-Language-Action Models from Egocentric Human Videos](https://arxiv.org/pdf/2507.12440)

[2] [Being-H0: Vision-Language-Action Pretraining from Large-Scale Human Videos](https://arxiv.org/pdf/2507.15597)

[3] [Deep Sensorimotor Control by Imitating Predictive Models of Human Motion](https://arxiv.org/pdf/2508.18691)

[4] [RT-Trajectory: Robotic Task Generalization via Hindsight Trajectory Sketches](https://arxiv.org/pdf/2311.01977)

[5] [TraceVLA: Visual trace prompting enhances spatial-temporal awareness for generalist robotic policies](https://arxiv.org/pdf/2412.10345)

[6] [Ego4D: Around the World in 3,000 Hours of Egocentric Video](https://arxiv.org/pdf/2110.07058)

[7] [Scaling Egocentric Vision: The EPIC-KITCHENS Dataset](https://arxiv.org/abs/1804.02748)

**Questions:**

In addition to the weaknesses above:

1.	Can the authors provide an analysis of the failure cases of SVA?

2.	Can the approach be validated on real robots to follow traces?

3.	For human-drawn steerable traces, what may influence trace-following performance?

4.	What explains the large discrepancy between Fig. 7 and Fig. 3 in SVA’s reported FVD scores?

---

### Official Review · Reviewer_CMAa · 2025-10-27

**Soundness:** 2
**Presentation:** 3
**Contribution:** 2
**Rating:** 2
**Confidence:** 3

**Summary:**

This paper introduces the Steerable Video Action (SVA) model, a unified video-action architecture that conditions on end-effector trajectory traces. This formulation enjoys the benefits of the video action models while additionally enabling the model to be effectively steered with traces in unfamiliar scenarios. The method supports joint pretraining with human video data. Experiments are conducted on the LIBERO benchmark and show improvements over baselines on trace-following and generalization to unseen traces.

**Strengths:**

- Proposed an effective and novel trace-following video action model.
- The steering signal(traces), can serve as a guidance in unseen scenarios where similar methods would normally fail.
- Supports pretraining with actionless videos (A common strength for video action models).
- Showed that the spatial alignment technique is crucial for accurate trace following.

**Weaknesses:**

## A. Trace Availability & Real-World Feasibility

- The method requires additional trajectory traces as input, but the authors did not address the source of such traces. While being able to control the policy output is an intriguing feature, in practice, it can be hard to automatically obtain such trajectory traces.

- The current source of training trajectories in simulation relies on GT robot poses provided by the simulator. As the method currently only has simulated experiments, it is unclear how the method can be deployed in realistic scenarios where GT robot poses are not available. This is especially important since the authors claimed as a contribution that the framework can be trained with actionless videos, not just human videos, where traces can be generated with off-the-shelf methods.

## B. Baselines & Fairness of Comparison

- In Figure 5, the trace-free non-DP baseline (UVA) actually outperforms the steerable DP baseline (DP-Tr) by a noticeable margin, which seems to suggest that the dominant factor driving the improvement in this figure is the video-prediction objective rather than the proposed trace-following mechanism. This raises a concern that the DP-based baselines may be too weak (i.e., even with traces, they perform significantly worse than UVA). As the DP baselines are the only baselines conditioned on traces, the reviewer strongly recommends that the authors provide more explanation or analysis regarding this point.

- The stronger baselines (UVA) used in the main experiments did not use trace information, which makes the comparison in Figure 5 unfair. The authors should put more emphasis on this, as the reviewer believes that having traces as input is quite a big assumption.


## C. Missing Validation of Claimed Benefits

- It would be nice if the authors could show that joint training, along with human/other data, brings more improvement. Looking at the error bars in Figure 6, the current ~3% gain seems marginal, which makes it hard to assess whether the proposed joint training strategy actually helped.

- In the Conclusion, the authors mentioned that joint training along with traces can potentially enable richer latent space learning. It would benefit the work if the authors could provide additional experiments on this basis — e.g., benchmarking the proposed method pretrained with traces but tested without conditioning on traces during test-time (unconditional generation).


## D. Scope & Reproducibility

- The method was only tested on the LIBERO benchmark. It would be more convincing to include one or two additional environments, potentially with different embodiments for joint learning. This would not only strengthen the solidity of the method but also better support the claimed strength of cross-embodiment learning.

- The current version of the paper lacks technical details (appendix/supplementary materials). As the method section only introduces the high-level concept of the proposed framework, it is very hard to reproduce the claimed results. The reviewer would urge the authors to include such details (e.g., key model hyperparameters, training hardware/cost, inference latency).

**Questions:**

1. How do the authors envision acquiring reliable trajectory traces at deployment time, especially outside simulation, where GT robot poses are unavailable?

2. Since UVA (trace-free) outperforms DP-Tr (trace-based) in Figure 5, the paper seems to be missing a strong trace-based baseline. How would the authors justify that the current baseline setup is fair?

3. Can the authors expand experimental scope beyond LIBERO and provide more technical details (training/inference setup, dataset specs) to improve reproducibility and support generalization claims?

---

### Official Review · Reviewer_mBzK · 2025-11-01

**Soundness:** 3
**Presentation:** 3
**Contribution:** 2
**Rating:** 2
**Confidence:** 4

**Summary:**

The authors propose a steerable video action (SVA) model that Learns joint video-action to predict sequences of future video frames and low-level robot actions, given steering input in the form of end-effector traces in 2D with time encoded as color of the trace.  The authors claim that this results in more accurate execution of the steering inputs - this is illustrated through simulation experiments.  The authors also show that SVA generalizes better to unseen traces by showing an improvement of 33% over baselines.  The proposed SVA also allows for pretraining with actionless data, where action-free video data is used to first train the model followed by finetuning with embodiment-specific action data.  The authors claim this improves steerability (following traces) and generalization (following unseen traces).

**Strengths:**

- Steers a policy through 2D end-effector trajectories (rather than language, sketches, keypoints)
- Enables co-training with actionless data by pretraining with action-free video data and then finetune with embodiment-specific action.  The trained model could be finetuned for different embodiments with different actions.
- Considers variants of trace encoding.

**Weaknesses:**

- The novelty in this paper seems to weak - it's just another way to take steering input to output actions.
- It appears like the viewpoint of the video and the traces need to be the same?  If so, this is a weakness of the method.
- The viewpoint for the video needs to be static - 3rd person view.  Would it be stronger if this could be video from a camera on the robot.

**Questions:**

- Can the viewpoint of the generated video differ from that of the input?
- Do the viewpoint of the video frames and the traces need to align, or can they be in different viewpoints?
- Unclear why trajectory and time are encoded as colored 2D traces.  Why not just provide a time sequence of end-effector positions?

---

### Note · Authors · 2026-01-21

**Comment:**

We thank the reviewers for their thoughtful and constructive feedback. Given the overall initial recommendations, we decided to withdraw this submission. We appreciate the reviewers’ and Area Chair’s time and effort.

**Withdrawal Confirmation:**

I have read and agree with the venue's withdrawal policy on behalf of myself and my co-authors.